# Towards Controllable Diffusion Models via Training-Phase Guided Exploration

## Abstract

By formulating data samples' formation as a Markov denoising process, diffusion models achieve state-of-the-art performances in a collection of tasks. Recently, many variants of diffusion models have been proposed to enable controlled sample generation. Most of these existing methods either formulate the controlling information as an input (i.e.,: conditional representation) for the noise approximator, or introduce a pre-trained classifier in the test-phase to guide the Langevin dynamic towards the conditional goal. However, the former line of methods only work when the controlling information can be formulated as conditional representations, while the latter requires the pre-trained guidance classifier to be differentiable. In this paper, we propose a novel framework named *RGDM* (**R**eward-**G**uided **D**iffusion **M**odel) that guides the training-phase of diffusion models via reinforcement learning (RL). The proposed training framework bridges the objective of weighted log-likelihood and maximum entropy RL, which enables calculating policy gradients via samples from a pay-off distribution proportional to exponential scaled rewards, rather than from policies themselves. Such a framework alleviates the high gradient variances and enables diffusion models to explore for highly rewarded samples in the reverse process. Experiments on 3D shape and molecule generation tasks show significant improvements over existing conditional diffusion models.

## 1 Introduction

Diffusion models have already shown their great success in density estimation Ho et al. (2020); Song & Ermon (2019); Nichol & Dhariwal (2021); Kingma et al. (2021), image synthesis Rombach et al. (2022); Ho et al. (2022), 3D shape generation Luo & Hu (2021); Zhou et al. (2021), audio synthesis Chen et al. (2020); Kong et al. (2020) and super-resolution Saharia et al. (2022b). This series of models define a Markov diffusion process that gradually adds random noise to data samples and then learns a reversed process to denoise the perturbations added in the diffusion process to reconstruct data samples from the noise. Ho et al. Ho et al. (2020) showed that diffusion models essentially learn gradients of data distribution density, which is equivalent to score-based generative models like Song & Ermon (2019). Recently, a collection of literature Luo & Hu (2021); Ho et al. (2020); Rombach et al. (2022); Dhariwal & Nichol (2021); Ho & Salimans (2022); Saharia et al. (2022a); Choi et al. (2021); Song et al. (2021) proposes multiple variants of diffusion models to enable more precise control of generation results. Such controlled models directly benefit a collection of commercial applications, such as STABLE Diffusion[1] and ERNIE-ViLG[2].

Based on the methodology, existing conditional diffusion models can be categorized into three types. The first type of works Luo & Hu (2021); Ho et al. (2020); Ho & Salimans (2022); Rombach et al. (2022) directly introduces conditional variables to construct conditional noise estimator for diffusion models. However, this line of methods can only be applied when conditional information can be formulated as representation variables. The second type of works Dhariwal & Nichol (2021) manipulates the generation results by introducing pre-trained classifiers Dhariwal & Nichol (2021). Nevertheless, Dhariwal & Nichol (2021) requires the pre-trained classifier to be differentiable. Conditional guidance from regression models or non-differentiable classifiers, such as random forest, cannot be used under Dhariwal & Nichol (2021). Thus, its application scope is also limited. Finally,

---

[1] https://huggingface.co/spaces/stabilityai/stable-diffusion
[2] https://huggingface.co/spaces/PaddlePaddle/ERNIE-ViLG

there are also some works design task-specific Saharia et al. (2022a); Choi et al. (2021); Song et al. (2021) (*e.g.* image-to-image translation or linear inverse imaging) conditional generation models. These methods typically require additional reference such as images, and can hardly precisely decide the generation outcome.

To tackle the drawbacks, we propose to guide the reversed process of a diffusion model via a reinforcement learning (RL) reward function for flexible and controllable generation. This is because the diffusion/reversed process in diffusion models and the Markov decision process (MDP) in RL both follow the Markov property. However, directly applying classic RL algorithms (*e.g.* policy gradient Sutton et al. (1999), Q-learning Mnih et al. (2013)) to train reward-guided diffusion models can lead to performance and efficiency issues. The reasons are as follows. First, unlike autonomous controlling, the length of reversed processes can be as long as hundreds of steps. Thus, the estimated gradient given such long episodes faces high variance. Moreover, in classic RL algorithms, gradients are calculated via episodes sampled directly from the policies themselves, which is costly and non-station. Last but not least, for some non-smooth reward functions, it is difficult for diffusion models to find any highly rewarded samples. Due to these discrepancies, there is still a noticeable gap before applying RL techniques to diffusion models in real practice.

In this paper we develop a novel training framework named *RGDM* for diffusion models. Specifically, the proposed *RGDM* draws intermediate samples in the reversed process from a reward-aware pay-off distribution instead of the estimated diffusion model; and utilizes these samples to compute policy gradients and update the diffusion model. Compared with traditional RL methods Sutton et al. (1999); Mnih et al. (2013; 2016), which rely on samples from the policy (model) to calculate gradients, the proposed framework not only reduces the variance of estimated gradients but also avoids expensive sampling from the non-stationary policy (model). The theoretical analysis in this paper reveals that sampling from a stationary pay-off distribution enjoys an identical optimal point to maximum entropy RL Peters & Schaal (2007); Norouzi et al. (2016); Kappen et al. (2012).

## 2 RELATED WORKS

**Conditional Diffusion Models** A collection of diffusion models has been designed for conditional generation. Based on the methodology, existing work can be classified into three categories. The first line of methods Ho et al. (2020); Luo & Hu (2021); Rombach et al. (2022) introduce conditional variables and construct conditional noise approximators (or score approximators) for diffusion models. Early works either directly introduce condition variables as an input of the noise approximators Ho et al. (2020), or introduce an encoder Luo & Hu (2021) to encode more complex conditional knowledge (e.g., reference image or text). Later, works like Rombach et al. (2022) first project samples from image space to low-dimensional latent space and then perform diffusion models on the latent space. In each reversed step, the conditional representation is mapped to the intermediate layers of the noise approximators via a cross-attention layer. Ho & Salimans (2022) jointly train a conditional and an unconditional diffusion model, and we combine the resulting conditional, and unconditional score estimates to attain a trade-off between sample quality and diversity similar to that obtained using classifier guidance. The second line of methods Dhariwal & Nichol (2021) manipulates the sampling phase of diffusion models to guide a trained model to generate samples that satisfy certain requirements. For instance, Dhariwal & Nichol (2021) proposes to utilize a pre-trained classifier $p(y \mid x_t)$ to guide the denoising model towards generating images with the attribute $y$.

There are also some works Saharia et al. (2022a); Choi et al. (2021); Song et al. (2021) design task-specific conditional generation methods for image-to-image translation and linear inverse imaging problems. For example, Saharia et al. Saharia et al. (2022a) directly injects a corrupted reference image into the noise approximators of diffusion models for image-to-image translation. Choi et al. Choi et al. (2021) proposes a method upon unconditional DDPM. Particularly, it introduces a reference image to influence the vanilla generation process. In each transition of the reversed process, the intermediate denoising result is synthesized with the corrupted reference image to let the reference image influence the generation result. Unlike the existing conditional diffusion model, this paper proposes a flexible reinforced framework that utilizes a pre-trained classifier/regressor to guide the diffusion model towards the desired condition in the training phase. To our knowledge, this is the first training-phase conditional diffusion framework alternative to directly introducing condition variables.

**3D Point Cloud Generation** This paper is also related to 3D shape generation since the experiments are carried out on three point cloud datasets. Shape generation via deep model is a fundamental topic in computer vision. Unlike images, 3D shapes can be represented as: voxel grids, point clouds and meshes, etc. In this paper, we focus on generating 3D shapes in the form of point clouds from scratch. Most existing works in this domain can be roughly classified into: *autoregressive-based* Sun et al. (2020) (learn the joint distribution of 3D point coordinates, and sample points one-by-one in the generation phase), *flow-based* Yang et al. (2019); Mittal et al. (2022); Klokov et al. (2020); Cai et al. (2020) (learn a sequence of invertible transformations of points that can transform real data to a simple distribution or vice-versa.) and *GAN-based* Shu et al. (2019); Li et al. (2021); Ramasinghe et al. (2020); Tang et al. (2022) (simultaneously learn the generator and the discriminator via a mini-max game). Recently, Luo et al. Luo & Hu (2021) adopts a diffusion model to learn a Markov reverse diffusion process for point clouds given a latent distribution. Similar idea is also used in existing score-based approaches, such as ShapeGF Cai et al. (2020). Essentially, these two methods learn the gradient density of data distribution and gradually move points along gradients. *We have included Luo & Hu (2021) in our experiment as a comparison method.*

Finally, we acknowledge that a recent work Decision Diffuser Ajay et al. (2022) also combines RL with diffusion models. However, this work differs significantly from Decision Diffuser. Decision Diffuser essentially adopts noise estimator as the policy of offline RL, which aims to improve offline RL algorithms with better exploration properties. This work, on the other hand, focuses on designing reward-aware exploration strategies for diffusion models, which can adaptively adjust the exploration according to the online reward feedback. In addition to the differences mentioned earlier, the technical approaches of these two papers are also completely different.

## 3 BACKGROUND: DIFFUSION MODELS

Considering a data sample $\mathbf{X}_0$, diffusion models such as Ho et al. (2020); Nichol & Dhariwal (2021); Kingma et al. (2021); Sohl-Dickstein et al. (2015) are inspired by non-equilibrium thermodynamics, in which data points gradually diffuse into chaos. A diffusion model consists of a forward *diffusion process* and a learnable backward *reversed process*.

**Diffusion Process** In the diffusion process, multivariate Gaussian noise is added to the sample step-by-step which is Markovian. The transition distribution from step $t-1$ to step $t$ is formulated as: $q(\mathbf{X}_t \mid \mathbf{X}_{t-1}) = \mathcal{N}(\mathbf{X}_t; \alpha_t \mathbf{X}_{t-1}, \beta_t \mathbf{I})$, where $\mathbf{X}_t$ denotes the noisy intermediate sample in step $t$ and $\mathbf{I}$ is an identity matrix. Two sets of noise schedule parameters, $\alpha_t$ and $\beta_t$, control how much signal is retained, and how much noise is added, respectively. Those parameters are either formulated in fixed form Sohl-Dickstein et al. (2015); Ho et al. (2020) or learned via neural networks Kingma et al. (2021). In this paper, we adopt *variance-preserving* strategy as used by Ho et al. (2020); Sohl-Dickstein et al. (2015) to let $\alpha_t = 1 - \beta_t$.

**Reversed Process** The reversed process, which is viewed as the reverse of the diffusion process, aims to generate meaningful samples from random noise. In such a process, the random noise reversely passes through the Markov chain in the diffusion process and recovers the desired sample.

Suppose we know the exact reverse distribution as $q(\mathbf{X}_{t-1} \mid \mathbf{X}_t, \mathbf{X}_0)$, we can reversely run the diffusion process to get a denoised sample Ho et al. (2020); Kingma et al. (2021); Nichol & Dhariwal (2021). However, $q(\mathbf{X}_{t-1} \mid \mathbf{X}_t, \mathbf{X}_0)$ is inducted based on the entire data distribution. Hence, we approximate it using a neural network: $p_\theta(\mathbf{X}_{t-1} \mid \mathbf{X}_t) = (\mathbf{X}_{t-1} \mid \boldsymbol{\mu}_\theta(\mathbf{X}_t, t), \eta_t \mathbf{I})$, where $\eta_t$ is the scheduled variance at step $t$ and we set $\eta_t = \beta_t$. $\boldsymbol{\mu}_\theta(\mathbf{X}_t, t)$ is a learnable estimator for the mean of $\mathbf{X}_{t-1}$ w.r.t $\mathbf{X}_t$ and $t$. For better description, we postpone the detailed implementation of $\boldsymbol{\mu}_\theta(\mathbf{X}_t, t)$ and $\eta_t$ in *the next paragraph*.

**Training** The training of diffusion models is performed by minimizing the variational lower bound on negative log likelihood:

$$\mathcal{L}_{MLE} = \mathbb{E}(-\log p(\mathbf{X}_0))$$

$$\leq \underbrace{D_{\mathrm{KL}}(q(\mathbf{X}_T \mid \mathbf{X}_0) \parallel p(\mathbf{X}_T))}_{\mathcal{L}_T} + \sum_{t=1}^{T} \underbrace{D_{\mathrm{KL}}(q(\mathbf{X}_{t-1} \mid \mathbf{X}_t, \mathbf{X}_0) \parallel p_\theta(\mathbf{X}_{t-1} \mid \mathbf{X}_t))}_{\mathcal{L}_{ll,t}}. \quad (1)$$

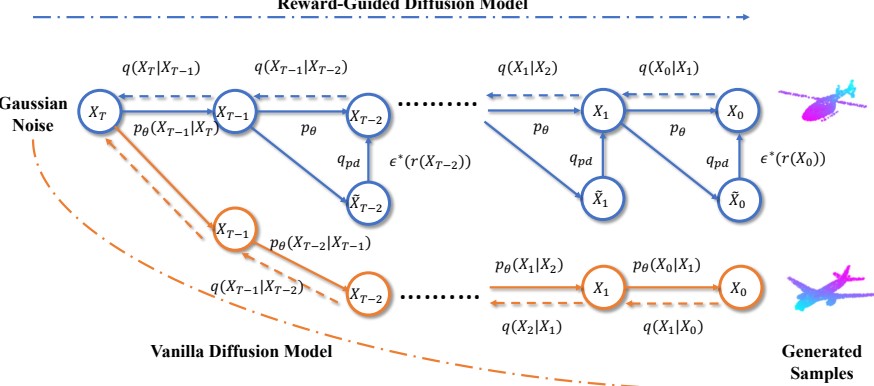

Figure 1: The directed graphical model of the proposed *RGDM* . The orange and blue trajectory denote the diffusion/reverse process of vanilla DDPM and *RGDM* , respectively.

Here, $\mathcal{L}_{ll,t}$ suggests that the estimated reverse distribution $p_\theta(\mathbf{X}_{t-1} \mid \mathbf{X}_t)$ should be close to the exact reverse distribution $q(\mathbf{X}_{t-1} \mid \mathbf{X}_t, \mathbf{X}_0)$. $\mathcal{L}_T$ characterizes the deviation between $q(\mathbf{X}_T \mid \mathbf{X}_0)$ and a standard Gaussian $p(\mathbf{X}_T)$. Here $q(\mathbf{X}_{t-1} \mid \mathbf{X}_t, \mathbf{X}_0)$ is formulated as: $q(\mathbf{X}_{t-1} \mid \mathbf{X}_t, \mathbf{X}_0) = \mathcal{N}(\mathbf{X}_{t-1} \mid \boldsymbol{\mu}_t(\mathbf{X}_t, \mathbf{X}_0), \gamma_t \mathbf{I})$, where $\gamma_t = \frac{\beta_t(1-\alpha_{t-1})}{\bar{\alpha}_t}$. and $\boldsymbol{\mu}_t(\mathbf{X}_t, \mathbf{X}_0) = \frac{\sqrt{\alpha_t}(1-\bar{\alpha}_{t-1})}{1-\bar{\alpha}_t}\mathbf{X}_t + \frac{\beta_t\sqrt{\bar{\alpha}_{t-1}}}{1-\bar{\alpha}_t}\mathbf{X}_0$ with $\bar{\alpha}_t = \prod_{m=1}^{t}\alpha_m$. In Ho et al. (2020), $\mu_\theta(\mathbf{X}_t, t)$ is parameterized as $\boldsymbol{\mu}_\theta(\mathbf{X}_t, t) = \frac{1}{\sqrt{\alpha_t}}\mathbf{X}_t - \frac{\beta_t}{\sqrt{\alpha_t(1-\bar{\alpha}_t)}}\boldsymbol{\epsilon}_\theta(\mathbf{X}_t, t)$. With both $p_\theta(\cdot)$ and $q(\cdot)$ in Gaussian, one can write the KL divergence of $\mathcal{L}_{ll,t}$ in closed-form, as suggested in Ho et al. (2020) and Kingma et al. (2021):

$$\mathcal{L}_{ll,t} = \mathbb{E}_q \frac{1}{2\eta_t}||\boldsymbol{\mu}_t(\mathbf{X}_t, \mathbf{X}_0) - \boldsymbol{\mu}_\theta(\mathbf{X}_t, t)||^2 = \mathbb{E}_q \frac{1}{2\eta_t}||\boldsymbol{\epsilon} - \boldsymbol{\epsilon}_\theta(\mathbf{X}_t, t)||^2. \tag{2}$$

**Sampling** The sampling process resembles Langevin dynamics with the estimator of data density gradients, i.e., $\boldsymbol{\epsilon}_\theta$. Concretely, the sampling process generates intermediate samples iteratively via:

$$\widehat{\mathbf{X}}_{t-1} = \frac{1}{\sqrt{\alpha_t}}\mathbf{X}_t - \frac{\beta_t\boldsymbol{\epsilon}_\theta(\mathbf{X}_t, t)}{\sqrt{\alpha_t(1-\bar{\alpha}_t)}} + \sqrt{\beta_t}\boldsymbol{\epsilon}, \forall t = T, \cdots, 1. \tag{3}$$

## 4 *RGDM* : REWARD GUIDED DIFFUSION MODEL

### 4.1 OVERVIEW

Applying RL to diffusion model training is quite intuitive, since diffusion models perform generation via iterative refinements, which can be viewed as a Markov multi-step decision process in the context of RL. Concretely, given a dataset $D$, we compute $\mathbb{E}_D(r(\mathbf{X}))$ as a measure of the empirical reward, which evaluates the fulfillment of the controllable generation goal, and hopes to maximize empirical rewards during training.

In this paper, we maximize the expected reward with a maximum entropy regularizer Haarnoja (2018), given as: $\max_\pi \mathbb{E}_D \sum_t [r_t + \mathcal{H}(\pi(\cdot|s_t))]$, where $r, s, \pi, \mathcal{H}$ denote reward, state, policy and entropy, respectively. By treating $p_\theta(\mathbf{X}_{t-1} \mid \mathbf{X}_t)$ in diffusion models as RL policy, one can adopt such a learning objective to train reward-guided diffusion models:

$$\mathcal{L}_{RL} = -\mathbb{E}_D\left\{ \sum_{t=1}^{T} \left[ \mathcal{H}(p_\theta(\mathbf{X}_{t-1} \mid \mathbf{X}_t)) + \mathbb{E}_{p_\theta(\mathbf{X}_{t-1}|\mathbf{X}_t)} r(\mathbf{X}_{t-1}) \right] + \left[ \mathcal{H}(p_\theta(\mathbf{X}_0)) + \mathbb{E}_{p_\theta(\mathbf{X}_0)} r(\mathbf{X}_0) \right] \right\}, \tag{4}$$

where $r(\mathbf{X})$ refers to the reward function and $\mathcal{H}(p)$ denotes the entropy of a distribution $p$ (*i.e.* $\mathcal{H}(p(\mathbf{X})) = \int p(\mathbf{X}) \log p(\mathbf{X}) d\mathbf{X}$). The last two terms are constants since $\mathbb{E}_{p_\theta(\mathbf{X}_0)}$ and $p_\theta(\mathbf{X}_0)$ are constants for a fixed dataset. *We drop them in practice.*

Although Eq. (4) seems like a natural formulation, direct policy optimization faces significant challenges. First, diffusion models require hundreds of iterative refinements to generate high-quality samples. However, estimating gradients by directly sampling such long decision trajectories from the policy distribution is costly, non-station, and may suffer large variances. Moreover, for some non-smooth reward functions in a high-dimensional output space, it is difficult for learners to find any highly rewarded samples, especially in early diffusion / reverse steps.

To tackle these challenges, we propose a novel reward-guided framework, which lies upon the typical entropy regularized reward expectation objective (*i.e.* Eq. (4)), for diffusion model training. Particularly, we estimate the gradient of the objective (*i.e.* Eq. (4)) w.r.t. policy parameters by sampling from an exponential reward-aware payoff distribution rather than the policy itself. Our analysis suggests that by introducing the reward-aware payoff distribution, the entropy regularized reward expectation objective is naturally transformed into *reward re-weighted biased noise prediction loss* for diffusion models.

## 4.2 REWARD GUIDED SAMPLING

As mentioned above, optimizing $\mathcal{L}_{RL}$ using SGD is challenging. Motivated by Norouzi et al. (2016), we introduce a *exponential payoff distribution* $q_{pd}$ which links Eq. (1) and RL objectives:

$$\mathbf{X}_{t-1} \sim q_{pd}(\mathbf{X}_{t-1} \mid \mathbf{X}_t, \mathbf{X}_0) = \int p(\mathbf{X}_{t-1} \mid \tilde{\mathbf{X}}_{t-1}) q(\tilde{\mathbf{X}}_{t-1} \mid \mathbf{X}_t, \mathbf{X}_0) \mathrm{d}\tilde{\mathbf{X}}_{t-1}, \tag{5}$$

where:

$$\tilde{\mathbf{X}}_{t-1} \sim q(\tilde{\mathbf{X}}_{t-1} \mid \mathbf{X}_t, \mathbf{X}_0)$$

$$p(\mathbf{X}_{t-1} \mid \tilde{\mathbf{X}}_{t-1}) = \frac{1}{Z}[\exp(r(\mathbf{X}_{t-1})) \cdot \mathcal{U}(\tilde{\mathbf{X}}_{t-1}, d)], \text{ and } \mathcal{U}(\tilde{\mathbf{X}}_{t-1}, d) = \begin{cases} \frac{1}{\pi d} & \text{if } \|\mathbf{X} - \mathbf{X}_{t-1}\|^2 < d, \\ 0 & \text{otherwise.} \end{cases} \tag{6}$$

Here, $Z = \int p(\mathbf{X}_{t-1} \mid \tilde{\mathbf{X}}_{t-1}) \mathrm{d}\mathbf{X}_{t-1}$ is a constant. Distribution $q$ is identical to that in DDPM Ho et al. (2020). $\mathcal{U}(\tilde{\mathbf{X}}_{t-1}, d)$ is a 'restriction' distribution to force $\tilde{\mathbf{X}}_{t-1}$ to be adjacent to $\mathbf{X}_{t-1}$, *i.e.* within radius $d$. We refer to $d$ as *exploration radius*. Intuitively, in diffusion dynamics, intermediate results should move smoothly towards the desired shape (with high reward). The distance between $\mathbf{X}_{t-1}$ and $\tilde{\mathbf{X}}_{t-1}$ should be small. Given such a distribution, we can sample $\mathbf{X}_{t-1}$ by simply searching $\tilde{\mathbf{X}}_{t-1}$'s adjacent region for $\mathbf{X}_{t-1}$s with highest rewards in samples. Such a searching procedure can be viewed as exploring for the high-rewarded modifications in the diffusion/reversed process from the view of RL. Concretely,

$$\mathbf{X}_{t-1} = \tilde{\mathbf{X}}_{t-1} + \boldsymbol{\epsilon}_*(r(\mathbf{X}_{t-1})), \tag{7}$$

where $\boldsymbol{\epsilon}_*(r(\mathbf{X}_{t-1}))$ denotes the feasible shift on $\tilde{\mathbf{X}}_{t-1}$ to maximize $r(\mathbf{X}_{t-1})$.

Note that we do not make an assumption on the differentiability of $r(\mathbf{X}_{t-1})$. Essentially, $r(\mathbf{X}_{t-1})$ is treated as a blackbox. When $r(\mathbf{X}_{t-1})$ is differentiable, we solve for $\boldsymbol{\epsilon}_*(r(\mathbf{X}_{t-1}))$ via gradient-based approaches. Otherwise, one may use non-gradient optimization methods like simulated annealing to obtain $\boldsymbol{\epsilon}_*(r(\mathbf{X}_{t-1}))$.

## 4.3 REWARD GUIDED LOSS

One can verify that the global minimum of $\mathcal{L}_{RL}$, *i.e.* the optimal regularized expected reward, is achieved when the model distribution $p_\theta$ perfectly matches the exponentiated payoff distribution $q_{pd}$. To see this, we re-express the objective function in Eq. (4) in terms of a KL divergence between $p_\theta(\mathbf{X}_{t-1} \mid \mathbf{X}_t)$ and $q_{pd}(\mathbf{X}_{t-1} \mid \mathbf{X}_t, \mathbf{X}_0)$):

$$\mathcal{L}_{RL} \propto \mathbb{E}_D \sum_{t=1}^{T} D_{\mathrm{KL}}(p_\theta(\mathbf{X}_{t-1} \mid \mathbf{X}_t) \,\|\, q_{pd}(\mathbf{X}_{t-1} \mid \mathbf{X}_t, \mathbf{X}_0)). \tag{8}$$

Due to space limitation, we postpone the detailed derivation from Eq. (4) Eq. (8) to Appendix A.1.

Since the global minimum of Eq. (8) is achieved when $p_\theta(\mathbf{X}_{t-1} \mid \mathbf{X}_t)$ matches $q_{pd}(\mathbf{X}_{t-1} \mid \mathbf{X}_t)$, we can swap $p_\theta(\mathbf{X}_{t-1} \mid \mathbf{X}_t)$ and $q_{pd}(\mathbf{X}_{t-1} \mid \mathbf{X}_t)$ in Eq. (8), without impacting the learning objective:

$$\mathcal{L}_{RMLE} \propto \mathbb{E}_D \sum_{t=1}^{T} D_{\text{KL}}(q_{pd}(\mathbf{X}_{t-1} \mid \mathbf{X}_t, \mathbf{X}_0) \,\|\, p_\theta(\mathbf{X}_{t-1} \mid \mathbf{X}_t)). \tag{9}$$

The objective functions $\mathcal{L}_{RL}$ and $\mathcal{L}_{RMLE}$, have the same global optimum of $p$, but they optimize a KL divergence in opposite directions. When optimizing $\mathcal{L}_{RMLE}$, one can draw unbiased samples from the stationary exponential payoff distribution $q_{pd}$ instead of the model $p$ itself.

From Eq. (9), we derive the concrete learning objective:

$$\min_\theta \mathbb{E}_D \left\{ \sum_{t=1}^{T} \int q_{pd}(\log q_{pd} - \log p_\theta) \mathrm{d}\mathbf{X}_{t-1} \right\} \propto \mathbb{E}_D \left\{ \sum_{t=1}^{T} \mathbb{E}_{\mathbf{X}_t} \exp(r(\mathbf{X}_{t-1})) \frac{\|\mathbf{X}_{t-1} - \boldsymbol{\mu}_\theta(\mathbf{X}_t, t)\|^2}{2\beta_t} \right\}, \tag{10}$$

where we use $q_{pd}$ and $p$ to denote $q_{pd}(\mathbf{X}_{t-1} \mid \mathbf{X}_t, \mathbf{X}_0)$ and $p_\theta(\mathbf{X}_{t-1} \mid \mathbf{X}_t)$ for abbreviation. Please refer to Appendix for the detailed derivations.

To implement Eq. (10),, we reformulate $\mathbf{X}_{t-1}$ and $\boldsymbol{\mu}_\theta(\mathbf{X}_t, t)$ in Eq. (10) following Sec. 3 and Eq. (5) as:

$$\mathbf{X}_t = \sqrt{\bar{\alpha}_t}\mathbf{X}_0 + \sqrt{1 - \bar{\alpha}_t}\boldsymbol{\epsilon} \qquad \tilde{\mathbf{X}}_{t-1} = \frac{\sqrt{\bar{\alpha}_{t-1}}\beta_t}{1 - \bar{\alpha}_t}\mathbf{X}_0 + \frac{\sqrt{\alpha_t}(1 - \bar{\alpha}_{t-1})}{1 - \bar{\alpha}_t}\mathbf{X}_t$$

$$\mathbf{X}_{t-1} \sim q_{pd}(\mathbf{X}_{t-1} \mid \mathbf{X}_t, \mathbf{X}_0) = \tilde{\mathbf{X}}_{t-1} + \boldsymbol{\epsilon}_*(r(\mathbf{X}_{t-1})) \tag{11}$$

$$\boldsymbol{\mu}_\theta(\mathbf{X}_t, t) = \frac{1}{\sqrt{\alpha_t}}\mathbf{X}_t - \frac{\beta_t \boldsymbol{\epsilon}_\theta(\mathbf{X}_t, t)}{\sqrt{\alpha_t(1 - \bar{\alpha}_t)}},$$

where $\boldsymbol{\epsilon}_*(r(\mathbf{X}_{t-1}))$ denotes the shifting vector suggested by the pay-off distribution.

To this point, we have both $\mathbf{X}_{t-1}$ and $\mu_\theta(\mathbf{X}_t, t)$ estimated. Thus, we can rewrite Eq. (10) as a compact formulation:

$$\min_\theta \mathbb{E}_D \left\{ \sum_{t=1}^{T} -\mathbb{E}_{\mathbf{X}_t} \frac{r(\mathbf{X}_{t-1})}{2\beta_t} \left\| \boldsymbol{\epsilon}_*(r(\mathbf{X}_{t-1})) + \frac{\beta_t}{\sqrt{\alpha_t(1 - \bar{\alpha}_t)}} (\boldsymbol{\epsilon} - \boldsymbol{\epsilon}_\theta(\mathbf{X}_t, t)) \right\|^2 \right\}. \tag{12}$$

This is obtained by simply substitute $\mathbf{X}_{t-1}$ and $\mu_\theta(\mathbf{X}_t, t)$ in Eq. (10) according to Eq. (11).

**Remark** Compared to the learning objective of the vanilla DDPM (Eq. (2)), the reward-guided diffusion model (Eq. (10) or (12)) suggests a *reward re-weighted biased noise prediction loss*, which favors (1) the samples with high rewards and (2) critic changes in the reversed process of diffusion models that lead to high rewards. Besides, unlike conventional RL objective (Eq. (4)), in Eq. (12) all the sampled terms (*i.e.* $\boldsymbol{\epsilon}$, $\boldsymbol{\epsilon}_*(r(\mathbf{X}_{t-1}))$ and $\mathbf{X}_{t-1}$) are drawn from stationary distribution (Gaussian or the proposed payoff distribution) instead of the intermediate policy. Such a sampling strategy reduces the variances of estimated policy gradients, and pilots the reversed process of diffusion models toward more highly-rewarded directions.

**Training and Sampling** Since the reward is often sparse in a high-dimensional output space, 'smart' model initialization (pre-training) instead of random initialization is needed. That is to say, at the beginning of the training, we use the maximum likelihood estimation (MLE) to pre-train the diffusion model on the training set $D$. Then we use the proposed *RGDM* to adjust the diffusion model according to the guidance of the reward function. We summarize the learning algorithm in Alg. 1 in the Appendix (due to space limitation). The sampling process of the proposed *RGDM* is identical to vanilla diffusion models.

## 5 EXPERIMENTS

In this section, we report the experimental results on multiple benchmark data sets crossing two tasks, *i.e.* 3D shape generation and molecule generation. These two tasks represents the cases that rewards guidances are (1) "classifier-like" and differentiable (3D shape generation); and (2) non-differentiable (molecule generation).

## 5.1 Controlled 3D Shape Generation

**Dataset** We adopt a fine-grained 3D shape dataset named FG3D Liu et al. (2021), which is built upon the well-known 3D shape datasets such as ShapeNet Chang et al. (2015), Yobi3D https://www.yobi3d.com and 3D Warehouse Goldfeder & Allen (2008). FG3D contains 25,552 shapes from three general categories (Airplane, Car and Chair), which are further labeled into 13, 20 and 33 sub-categories. Please refer to Appendix C.1 for detailed data processing procedures.

**Evaluation Metrics** The generation task in this section is to force the generator to generate samples of a specific sub-category. We expect the generation samples to be as similar to real samples from the targeted sub-category as possible. Here, we choose sub-categories 'helicopter', 'bus' and 'bar (chair)' from general categories Airplane, Car and Chair, respectively.

We adopt three commonly used metrics in existing 3D shape generation literature Shu et al. (2019); Tang et al. (2022); Cai et al. (2020), *i.e. Minimum Matching Distance (MMD)*: (the averaged distance between generated shapes and shapes in the reference set); *Coverage (COV)*: (the fraction of shapes that can be matched as a nearest shape with the generated shape); *Jenson-Shannon divergence (JSD)*: (the distance between the generated set and the groundtruth). Good methods should have a low MMD, high COV and low JSD. For distance evaluation metrics in MMD and COV, we adopt both Chamfer distance (CD) and Earth Mover distance (EMD). Besides, following existing papers Yang et al. (2019); Luo & Hu (2021); Cai et al. (2020), we normalize both generated point clouds and groundtruth references into a bounding box of [-1, 1] to focus on shape rather than scale.

**Baselines** We evaluate the shape generation capability of our method against two representative state-of-the-art controllable diffusion models, including 3D-DDPM Luo & Hu (2021) (*introduce categorical conditional variable for controlled generation*) and Test-Phase Manipulation (TPM) Dhariwal & Nichol (2021) (*manipulate the sampling phase via a pre-trained classifier/regressor*). The descriptions of these baselines are already detailed in Sec. 2. For both the baselines and the proposed method, we use the neural network architecture in Luo & Hu (2021) as backbone of the diffusion model. Besides, in order to show that the proposed *RGDM* CANNOT be simply viewed as re-weighting the samples in the training set, we introduce a variant of DDPM by simply re-weight the samples $\mathbf{X}_0$ in the training set with $r(\mathbf{X}_0)$. Such a variant is named after Reweighted-DDPM.

**Implementation Details** For all the experiments in this task, we set the number of steps, *i.e.* $T$, in the diffusion process to 100. The noise scheduling factor of step $t$, *i.e.* $\beta_t$, is set to linearly increase from $\beta_1 = 0.0001$ to $\beta_T = 0.05$. We choose to implement the noise estimator $\epsilon_\theta$ as a 7-layer MLP with concatsquash Grathwohl et al. (2018) layers and LeakyReLU. The dimension of concatsquash layers are (3-128-256-512-256-128-3). The rewards in this task is calculated as: $r(\mathbf{X}_t) = \frac{f_k(\mathbf{X}_t)}{\sum_i f_i(\mathbf{X}_t)}$, where $f_k$ denotes prediction (probability) of a point cloud classifier for a given input $\mathbf{X}_t$ w.r.t. a given fine-grained class $k$ (*e.g.* helicopter). Here, we choose PointNet Qi et al. (2017) as the classifier.

### 5.1.1 Results Analysis

**Quantitative Results** Table 1 reports the quantitative results of the proposed method and all the baselines. From the results, we can see that the proposed *RGDM* consistently outperforms all the baselines for all the metrics by significant margins. Particularly, low MMDs suggest that the proposed *RGDM* enable generated shapes to be similar to the shapes within the desired sub-categories in terms of both the spatial and feature space. The high COVs suggest that our generated shapes have a good coverage of the shapes in the desired sub-categories.

By outperforming existing controllable diffusion model training frameworks, the proposed *RGDM* demonstrates its advantage in guiding diffusion models towards the desired sub-categories. As a comparison, gradients from pre-trained classifiers, which are used in the test-phase controlling method (i.e., TPM Dhariwal & Nichol (2021)), fail to provide enough guidance toward the desired sub-categories. Besides, despite outperforming TPM Dhariwal & Nichol (2021), contextual representations in (conditional) 3D-DDPM Luo & Hu (2021) cannot match the performance of *RGDM* . This is because (conditional) 3D-DDPM Luo & Hu (2021) treats conditional information and general shape structure information as disentangled, via separated representations. However, since shapes from different sub-categories still share common characteristics, the learned contextual representations

Table 1: Performance on the FG3D dataset. The classes in parentheses indicate the sub-category that we force the models to generate. CD, and EMD distances are multiplied by $10^1$.

| Shape | Model | MMD ↓ | | COV ( ↑) | | JSD ↓ |
|---|---|---|---|---|---|---|
| | | CD | EMD | CD | EMD | - |
| Airplane (helicopter) | 3D-DDPM | 0.176 | 1.642 | 34.29 | **42.86** | 0.2321 |
| | TPM | 0.244 | 1.772 | 30.32 | 38.17 | 0.2781 |
| | Reweighted-DDPM | 0.162 | 1.591 | 38.33 | **42.86** | 0.1993 |
| | *RGDM* | **0.113** | **1.340** | **51.43** | **42.86** | **0.1496** |
| Car (bus) | 3D-DDPM | 0.101 | 1.150 | 16.00 | 11.50 | 0.2326 |
| | TPM | 0.129 | 1.762 | 14.38 | 10.20 | 0.2471 |
| | Reweighted-DDPM | 0.083 | 1.097 | 42.23 | 32.70 | 0.1394 |
| | *RGDM* | **0.068** | **0.938** | **48.00** | **40.00** | **0.0979** |
| Chair (bar) | 3D-DDPM | 0.474 | 2.682 | 31.11 | 42.22 | 0.1624 |
| | TPM | 0.505 | 2.933 | 29.46 | 41.33 | 0.1990 |
| | Reweighted-DDPM | 0.312 | 2.311 | 37.42 | 46.34 | 0.1263 |
| | *RGDM* | **0.253** | **2.091** | **57.78** | **57.78** | **0.1063** |

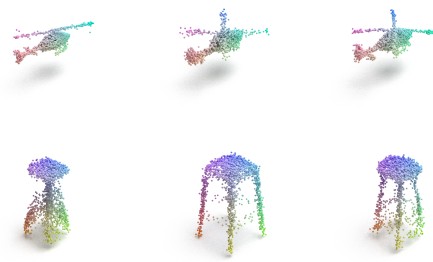

Figure 2: Generated point clouds with sub-categories, i.e., helicopter (airplane) and bar (chair).

Figure 3: Performance w.r.t. exploration radius $d$ under sub-category helicopter (airplane).

may not offer 'clean' enough guidance toward generating shapes of a specific sub-category. On the contrary, the proposed reward-guided objective in *RGDM* effectively forces the reversed trajectories toward the high-reward directions. That is why *RGDM* consistently outperforms 3D-DDPM Luo & Hu (2021). Finally, we witness the significant performance gap between Reweighted-DDPM in a vast majority of cases. Such an phenomenon indicates that that the proposed *RGDM* CANNOT be simply viewed as re-weighting the samples in the training set and the reward guided *RGDM* indeed helps the controlled generation.

**Effect of Exploration Radius** $d$   In the proposed framework, the exploration radius (*i.e.* $d$ in Eq. 6) is the only adjustable hyper-parameters that one can rely on (apart from noise scheduling hyper-parameters in vanilla DDPM) to influence the performance of *RGDM* . We illustrate the performance of *RGDM* w.r.t. parameter $d$ under the subcategory helicopter (airplane) in Figure 3. The tendency of other categories follows a similar tendency. It is evident that in most cases, choosing a value of $d$ greater than $0.5$ can result in improved performance. This observation suggests that adequate reward guided explorations can lead to better performance.

**Visualization**   We randomly pick several shape generated by *RGDM* given specific sub-categories (Figure 2). From these results, we can see that the point clouds generated by our method consists of fine details (e.g., propeller blade of helicopters) and little noise and few outlier points. The points in the shapes are also distributed uniformly, with no significant "holes".

## 5.2 CONTROLLED 3D MOLECULAR GENERATIONS

**Dataset**   QM9 (Ramakrishnan et al., 2014) is a dataset of 130k stable and synthetically accessible organic molecules with up to 9 heavy atoms (29 atoms including hydrogens). In this section, we train diffusion models to generate atoms' (1) 3-dimensional coordinates; (2) types (H, C, N, O, F) and (3) integer-valued atom charges. We use the train/val/test partitions introduced in Anderson et al. (2019) (train/val/test: 100K/18K/13K samples) for evaluation.

**Evaluation Metrics** Our goal here is to generate molecules targeting some desired properties while at the same time not harming general generation quality (e.g., molecules' validity (the proportion of atoms with right valency) and stability, etc.). In such a scenario, a molecule is represented as a point cloud, in which each point denotes a single atom and has its own (atom) type. Following Satorras et al. (2021a), for each pair of atoms, we use the distance between them and the atoms' types to predict bonds (single, double, triple, or none) between atoms.

In this section, we consider optimizing two desired properties: (1) quantitative estimate of drug-likeness (QED) Bickerton et al. (2012) (how likely a molecule is a potential drug candidate based on marketed drug molecules) and (2) synthetic accessibility score (SA) (the difficulty of drug synthesis), which are crucial in drug discovery domain. A good method should have a high averaged QED and SA. Note that we conduct separated experiments for these two properties, which means only one property is considered in a single experiment. We adopt widely-used open-source cheminformatics software RDKithttps://www.rdkit.org to calculate the properties above.

**Baselines** In this section, we adopt E(3) equivariant diffusion model (EDM) Hoogeboom et al. (2022), which is the most representative conditional molecule generation method, as the baseline. Due to space limitation, please refer to appendix for the details on the baseline method. For both the baseline and the proposed method, we use the noise estimator specified in Hoogeboom et al. (2022) (*i.e.* EGNNs with 256 hidden features and 9 layers) as the backbone.

**Implementation Details** For all the experiments in this section, we set the number of steps $T$ in the diffusion process to 1,000. The noise scheduling factor of step $t$, *i.e.* $\beta_t$, is set to the cosine noise schedule introduced in Nichol & Dhariwal (2021); Hoogeboom et al. (2022). The *rewards* in this task are defined as normalized QED and SA scores of the intermediate generated samples, *i.e.* $r_{QED}(\mathbf{X}_t) = \frac{QED(\mathbf{X}_t)}{\max_{\mathbf{X}_0 \sim D} QED(\mathbf{X}_0)}$ and, $r_{SA}(\mathbf{X}_t) = \frac{SA(\mathbf{X}_t)}{\max_{\mathbf{X}_0 \sim D} SA(\mathbf{X}_0)}$,, where $\max_{\mathbf{X}_0 \sim D} SA(\mathbf{X}_0)$ denotes the largest SA score of samples in the training set. The normalized QED and SA scores are from RDKit QED and SA calculator, i.e. function qed in [3], and function sascorer in [4].

### 5.2.1 RESULT ANALYSIS

The results of controlled molecule generation are reported in Table 2. Compared with EDM, the proposed reward-guided framework *RGDM* generates molecules with higher QED and SA scores without hurting their general chemistry characteristics (*i.e.* high rate of validity and stability). Such results again demonstrate the superiority of reward-guided diffusion models over existing conditional diffusion models, such as EDM.

Table 2: Molecule stability (Mol Stable), Validity (Valid), QED and SA across 3 runs on QM9, each drawing 1,000 samples.

| Methods | | EDM | *RGDM* |
|---|---|---|---|
| General Properties | Mol Stable (%) | 90.7 | 90.5 |
| | Valid (%) | 91.2 | 91.4 |
| Properties to Optimize | Avg. QED | 0.461 | **0.542** |
| | Avg. SA | 4.41 | **5.87** |

Moreover, it is worth emphasising that the molecular point clouds are much more sparse and discontinuous than 3D shape clouds. This characteristics makes it difficult for conventional RL learners to find highly-rewarded refinements in reversed trajectories. Nevertheless, by taking advantage of the reward-aware payoff distribution, the proposed *RGDM* successfully overcomes such difficulty and achieves impressive performance.

## 6 CONCLUSIONS

This paper presents a reward-guided learning framework for diffusion models that enables flexible controlled generation. Unlike the conventional RL framework, the proposed *RGDM* estimates policy (model) gradients through samples from the stationary reward-aware payoff distribution rather than the policy itself. Thus, the learning objective is turned into a *reward re-weighted biased noise prediction loss*, which can effectively guide the reversed process towards highly-rewarded directions and simultaneously reduce variances of the estimated gradients. Experimental results on 3D shape generation and molecular generation tasks show that the proposed framework outperforms existing controlled diffusion models by a clear margin.

---

[3]https://github.com/rdkit/rdkit/blob/master/rdkit/Chem/QED.py
[4]https://github.com/rdkit/rdkit/blob/master/Contrib/SA_Score/sascorer.py

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

## A APPENDIX I - DETAILED DERIVATIONS

### A.1 DERIVATION FROM EQ. (4) TO (8)

Below is the detailed derivation from Eq. (4) to (8):

$$\mathcal{L}_{RL}$$

$$= -\mathbb{E}_D\left\{\sum_{t=1}^{T}\left[\mathcal{H}(p_\theta(\mathbf{X}_{t-1}\mid\mathbf{X}_t)) + \mathbb{E}_{p_\theta(\mathbf{X}_{t-1}\mid\mathbf{X}_t)}r(\mathbf{X}_{t-1})\right]\right\}$$

$$= -\mathbb{E}_D\left\{\sum_{t=1}^{T}\left[\int p_\theta(\mathbf{X}_{t-1}\mid\mathbf{X}_t)\log p_\theta(\mathbf{X}_{t-1}\mid\mathbf{X}_t)\mathrm{d}\mathbf{X}_{t-1} + \int p_\theta(\mathbf{X}_{t-1}\mid\mathbf{X}_t)r(\mathbf{X}_{t-1})\mathrm{d}\mathbf{X}_{t-1}\right]\right\}$$

$$\propto -\mathbb{E}_D\left\{\sum_{t=1}^{T}\left[\int p_\theta(\mathbf{X}_{t-1}\mid\mathbf{X}_t)\log p_\theta(\mathbf{X}_{t-1}\mid\mathbf{X}_t)\mathrm{d}\mathbf{X}_{t-1} - p_\theta(\mathbf{X}_{t-1}\mid\mathbf{X}_t)\big(r(\mathbf{X}_{t-1})\right.\right.$$

$$\left.\left. \underbrace{- \log Z + \int q(\tilde{\mathbf{X}}_{t-1}\mid\mathbf{X}_t,\mathbf{X}_0)\mathrm{d}\tilde{\mathbf{X}}_{t-1}}_{Constants}\big)\mathrm{d}\mathbf{X}_{t-1}\right\}$$

$$\propto -\mathbb{E}_D\left\{\sum_{t=1}^{T}\left[\int p_\theta(\mathbf{X}_{t-1}\mid\mathbf{X}_t)\log p_\theta(\mathbf{X}_{t-1}\mid\mathbf{X}_t)\mathrm{d}\mathbf{X}_{t-1}\right.\right.$$

$$\left.\left. + \int p_\theta(\mathbf{X}_{t-1}\mid\mathbf{X}_t)\int p(\mathbf{X}_{t-1}\mid\tilde{\mathbf{X}}_{t-1})q(\tilde{\mathbf{X}}_{t-1}\mid\mathbf{X}_t,\mathbf{X}_0)\mathrm{d}\tilde{\mathbf{X}}_{t-1}\mathrm{d}\mathbf{X}_{t-1}\right]\right\}$$

$$= \mathbb{E}_D\left\{\sum_{t=1}^{T}\left[\int p_\theta(\mathbf{X}_{t-1}\mid\mathbf{X}_t)\Big(\log p_\theta(\mathbf{X}_{t-1}\mid\mathbf{X}_t) - \log q_{pd}(\mathbf{X}_{t-1}\mid\mathbf{X}_t,\mathbf{X}_0)\Big)\mathrm{d}\mathbf{X}_{t-1}\right]\right\}$$

$$= \mathbb{E}_D\sum_{t=1}^{T}\left\{\left[D_{\mathrm{KL}}(p_\theta(\mathbf{X}_{t-1}\mid\mathbf{X}_t)\,\|\,q_{pd}(\mathbf{X}_{t-1}\mid\mathbf{X}_t,\mathbf{X}_0))\right]\right\},$$

where $\int q(\tilde{\mathbf{X}}_{t-1}\mid\mathbf{X}_t,\mathbf{X}_0)\mathrm{d}\tilde{\mathbf{X}}_{t-1}$ and $\int p_\theta(\mathbf{X}_{t-1}\mid\mathbf{X}_t)\mathrm{d}\mathbf{X}$ are constants.

### A.2 DERIVATION OF EQ. (10)

Below is the detailed derivation of Eq. (10). Here, we introduce $\mathcal{F}(\mathbf{X}_{t-1})$, which denotes the feasible set of $\tilde{\mathbf{X}}_{t-1}$ given $\mathbf{X}_{t-1}$. In this paper, $\tilde{\mathbf{X}}_{t-1}$ lies within an $\ell_2$ ball of radius $d$ (as defined in Eq. (6)).

$$\mathcal{L}_{RMLE}$$

$$= \mathbb{E}\left\{\sum_{t=1}^{T}\int q_{pd}(\mathbf{X}_{t-1}\mid\mathbf{X}_t,\mathbf{X}_0)\big(\log q_{pd}(\mathbf{X}_{t-1}\mid\mathbf{X}_t,\mathbf{X}_0) - \log p_\theta(\mathbf{X}_{t-1}\mid\mathbf{X}_t)\big)\mathrm{d}\mathbf{X}_{t-1}\right\}$$

$$= \mathbb{E}\left\{\sum_{t=1}^{T}\int\left[\int_{\mathcal{F}(\mathbf{X}_{t-1})} p(\mathbf{X}_{t-1}\mid\tilde{\mathbf{X}}_{t-1})q(\tilde{\mathbf{X}}_{t-1}\mid\mathbf{X}_t,\mathbf{X}_0)\mathrm{d}\tilde{\mathbf{X}}_{t-1}\right.\right.$$

$$\times\Big(\log\int_{\mathcal{F}(\mathbf{X}_{t-1})} p(\mathbf{X}_{t-1}\mid\tilde{\mathbf{X}}_{t-1})q(\tilde{\mathbf{X}}_{t-1}\mid\mathbf{X}_t,\mathbf{X}_0)\mathrm{d}\tilde{\mathbf{X}}_{t-1}$$

$$\left.\left. - \frac{1}{2}\big(\log\beta_t - \log\pi - \log 2 - \frac{\|\mathbf{X}_{t-1}-\boldsymbol{\mu}_\theta(\mathbf{X}_t,t)\|^2}{\beta_t}\big)\Big)\right]\mathrm{d}\mathbf{X}_{t-1}\right\} \quad \Longleftarrow \frac{1}{Z}\exp(r(\mathbf{X}_{t-1}))\text{ does not include }\tilde{\mathbf{X}}_{t-1}.$$

$$= \mathbb{E}\left\{\sum_{t=1}^{T}\int\frac{1}{Z}\exp(r(\mathbf{X}_{t-1}))\left[\int_{\mathcal{F}(\mathbf{X}_{t-1})} q(\tilde{\mathbf{X}}_{t-1}\mid\mathbf{X}_t,\mathbf{X}_0)\mathrm{d}\tilde{\mathbf{X}}_{t-1}\right.\right.$$

$$\Big(\log\big(\frac{1}{Z}\exp(r(\mathbf{X}_{t-1}))\big)\int_{\mathcal{F}(\mathbf{X}_{t-1})} q(\tilde{\mathbf{X}}_{t-1}\mid\mathbf{X}_t,\mathbf{X}_0)\mathrm{d}\tilde{\mathbf{X}}_{t-1}\big)$$

$$\left.\left. - \frac{1}{2}\big(\log\beta_t - \log\pi - \log 2 - \frac{\|\mathbf{X}_{t-1}-\boldsymbol{\mu}_\theta(\mathbf{X}_t,t)\|^2}{\beta_t}\big)\Big)\right]\mathrm{d}\mathbf{X}_{t-1}\right\} \quad \Longleftarrow \text{finite integral of Gaussian family is a constant.}$$

$$
\begin{aligned}
=&\mathbb{E}\bigg\{\sum_{t=1}^{T}\int\frac{1}{Z}\exp(r(\mathbf{X}_{t-1}))\Big(r(\mathbf{X}_{t-1})-\log Z+\log\int_{\mathcal{F}(\mathbf{X}_{t-1})}q(\tilde{\mathbf{X}}_{t-1}\mid\mathbf{X}_t,\mathbf{X}_0)\mathrm{d}\tilde{\mathbf{X}}_{t-1}\\
&-\frac{1}{2}(\log\beta_t-\log\pi-\log 2-\frac{\|\mathbf{X}_{t-1}-\boldsymbol{\mu}_\theta(\mathbf{X}_t,t)\|^2}{\beta_t})\Big)\mathrm{d}\mathbf{X}_{t-1}\bigg\}\\
=&\mathbb{E}\bigg\{\sum_{t=1}^{T}\int\frac{1}{Z}\exp(r(\mathbf{X}_{t-1}))\Big(constant+\frac{\|\mathbf{X}_{t-1}-\boldsymbol{\mu}_\theta(\mathbf{X}_t,t)\|^2}{2\beta_t}\Big)\mathrm{d}\mathbf{X}_{t-1}\bigg\}\quad\Longleftarrow\text{ the formulation of }r(\mathbf{X}_{t-1})\text{ is unknown.}\\
\propto&\mathbb{E}\bigg\{\sum_{t=1}^{T}\mathbb{E}_{\mathbf{X}_t}\exp(r(\mathbf{X}_{t-1}))\frac{\|\mathbf{X}_{t-1}-\boldsymbol{\mu}_\theta(\mathbf{X}_t,t)\|^2}{2\beta_t}\bigg\}.
\end{aligned}
$$

## B  TRAINING ALGORITHM OF *RGDM* (ALGORITHM 1)

Algorithm 1 mentioned in Section 4.3 is as follows:

---
**Algorithm 1** *RGDM* Training Process
---
**Input:** Sample $\mathbf{X}_0$
1: Pre-train the diffusion model by optimizing Eq. (1).
2: **repeat**
3:    Sample $t\sim\mathcal{U}(0\cdots T)$.
4:    Calculate $\mathbf{X}_t$, $\mathbf{X}_{t-1}$ and $\boldsymbol{\mu}_\theta(\mathbf{X}_t,t)$ according to Eq. (11).
5:    Take a gradient descent step on Eq. (12).
6: **until** Converged
---

## C  EXTRA DETAILS IN EXPERIMENTS

### C.1  DATA PROCESSING PROCEDURE OF THE FG3D DATASET

FG3D Liu et al. (2021) is built upon the well-known 3D shape datasets such as ShapeNet Chang et al. (2015), Yobi3D https://www.yobi3d.com and 3D Warehouse Goldfeder & Allen (2008). FG3D contains 25,552 shapes from three general categories including Airplane, Car and Chair, which are further labeled into 13, 20 and 33 sub-categories. For each shape, we randomly sample 2048 points via Open3D[5] to obtain the point clouds and normalize the point clouds to zero mean and unit variance. Finally, we randomly split each sub-category into training, validation and testing sets by the ratio 90%, 5% and 5%, respectively.

### C.2  DETAILS ON THE BASELINE METHOD USED IN 3D MOLECULE GENERATION TASK

EDM specifies a diffusion process that operates on both continuous coordinates and categorical atom types. Particularly, EDM adopts equivariant graph neural networks (EGNN) Satorras et al. (2021b) to model molecular geometries, which are equivariant to the action of rotations, reflections and translations. A very recent work Bao et al. (2023) propose energy-aware guidance for molecule generation. However, it requires the energy-aware guidance to be differentiable, which is not directly applicable to the molecular property optimization tasks in this section. Besides, we exclude TPM Dhariwal & Nichol (2021) as a comparison method because it requires gradients from a pre-trained classifier (scoring function) to guide the controlled generation. Nevertheless, the property calculators for QED and SA are non-differentiable.

---
[5]www.open3d.org

