# OpenReview forum: "Towards Controllable Diffusion Models via Training-Phase Guided Exploration"
_ICLR.cc/2024/Conference — ICLR 2024 Conference Withdrawn Submission_

### Official Review · Reviewer_R9e8 · 2023-10-21

**Soundness:** 2 fair
**Presentation:** 2 fair
**Contribution:** 2 fair
**Rating:** 5
**Confidence:** 2

**Summary:**

The paper "Towards Controllable Diffusion Models via Training-Phase Guided Exploration" delves into the domain of diffusion models, which have exhibited remarkable success in a variety of tasks like density estimation, image synthesis, and 3D shape generation. While previous methods have attempted to steer sample generation through conditional inputs or the use of differentiable pre-trained classifiers, they come with their limitations. The authors introduce a new framework, RGDM (Reward-Guided Diffusion Model), that leverages reinforcement learning to guide the training phase of diffusion models.  The key innovation lies in drawing intermediate samples from a reward-aware pay-off distribution, as opposed to the model's estimates. This method, when tested on 3D shape and molecule generation tasks, demonstrated significant advancements over existing conditional diffusion models.

**Strengths:**

The concept of employing reinforcement learning (RL) to steer diffusion processes is both intuitive and ripe for investigation. The author skillfully establishes a linkage between the weighted log-likelihood objective and maximum entropy RL. This connection enables the efficient calculation of policy gradients through samples sourced from a reward-aware payoff distribution, which aligns proportionally with exponentially scaled rewards, rather than extracting directly from the policies themselves. The author posits that this methodology significantly diminishes gradient variance, thereby enhancing the diffusion models' capability to actively search for and identify samples with elevated rewards throughout the reverse process.

**Weaknesses:**

1. The author claims "Conditional guidance from regression models or non-differentiable classifiers, such as random forest,
cannot be used under Dhariwal & Nichol (2021). " I am having difficulty grasping this particular point. On the surface, integrating a regression model with the diffusion process seems to be a direct and uncomplicated task.

2. In the vicinity of Equation (7), the extensive use of underlining for emphasis detracts from the visual appeal of the document. To enhance the layout, the author could consider utilizing bold or italicized text for highlighting purposes.

3. The author claims “Since the global minimum of Eq. (11) is achieved when pθ(Xt−1 | Xt) matches qpd(Xt−1 | Xt), we
can swap pθ(Xt−1 | Xt) and qpd(Xt−1 | Xt) in Eq. (11), without impacting the learning objective” I am struggling to comprehend the rationale behind swapping terms in the equation mentioned, and I remain unconvinced of its validity. I believe the author needs to provide a solid justification for this action, as it seems to me that such a swap would indeed affect the learning objective.

4. I find myself curious about the author's choice to exclude experiments on well-known datasets such as CIFAR and ImageNet in the validation of their method's efficacy. The focus on 3D datasets raises questions, especially since the method in discussion is not uniquely tailored for 3D applications.

5. Pertaining to the "CONTROLLED 3D MOLECULAR GENERATIONS" experimental section, it appears to me that a straightforward and fundamental baseline could involve training a regression model and subsequently applying it directly to the diffusion generation process. This potential baseline seems to have been overlooked by the author.

**Questions:**

See Weaknesses.

---

### Official Review · Reviewer_ZNCD · 2023-10-31

**Soundness:** 3 good
**Presentation:** 3 good
**Contribution:** 3 good
**Rating:** 5
**Confidence:** 4

**Summary:**

The authors connect reward conditional sampling from a diffusion model and reinforcement learning. The result is modified training objective and sampling procedure that favors diffusion trajectories that enter high reward regions of the state space. In their experiments, the authors show that this improved method can generate 3d point clouds and molecules that sample from the desired conditional distribution (a desired discrete class, or maximizing a scalar value) while still maintaining overall plausibility and diversity of samples.

**Strengths:**

The idea is fairly natural but still interesting. The presentation is fairly clear overall, and the results seem promising.

**Weaknesses:**

It's not completely clear to me how the method is actually implemented. For each minibatch, is this the procedure as follows? (1) sample a data point (2) noise the data point (3) predict the denoised datapoint (4) sample the next step in the reverse process (5) score the sample and solve for the optimal feasible shift (7) calculate the loss.

Step (5) in this procedure seems potentially problematic to me. Not only is it potentially quite expensive for non-differentiable objectives (as the optimization needs to be performed at every training and sampling step for each point individually), but applying the reward model to intermediate samples might make very little sense. For example, in the case of small molecule structures and non-differentiable simulated reward functions, samples near the Gaussian prior often will not be meaningful molecules. How should the search be conducted over the local neighborhood of these samples (to optimize the reward) when we might not even be able to run the reward model in a meaningful way. It's often for this reason that people apply the guidance function to an estimate of x_0, the denoised state, or use an approximation of the reward function that is trained on noisy examples. Separately, in the case of generating class-conditional point clouds, how exactly does the training procedure work? Is the ground-truth label of each training data point used to obtain the correct class logit from the pretrained classifier? If so, it feels a bit odd to train one denoising network under this modified objective without passing the class label into the network, as the reward depends on the sampled class label.

In the molecular experiments, how is conditional generation being performed with EDM? I'm not familiar with them having QED results. Are you presenting completely unconditional samples from EDM or is there a learned, differentiable approximation to QED? It's possible that this comparison is not as strong as it could be and something like DiGress (Vignac et al 2022) might be another relevant baseline.

How is the exploration radius supposed to be chosen or tuned? The authors mention to 0.5 is a reliable setting, but in some cases performance is highly effected. It's not clear to me how to choose the right setting without potentially overfitting the dataset at hand.

In general, I'm also curious how the authors view the need for custom training per reward function with their proposed method. One perk of approaches that combine a pretrained unconditional generative model with a pretrained discriminative model is that arbitrary combinations can be created in a plug and play manner. Within this framework, a new model must be trained every time.

**Questions:**

The proposed method also seems potentially related to GFlowNets, and it might be helpful and/or interesting to hear the authors compare their approach with this family of methods.

---

### Official Review · Reviewer_TeCp · 2023-10-31

**Soundness:** 1 poor
**Presentation:** 2 fair
**Contribution:** 2 fair
**Rating:** 3
**Confidence:** 5

**Summary:**

This paper describes a framework called Reward-Guided Diffusion Model (RGDM) that guides the reverse process of diffusion models using a reward function. The key idea behind RGDM is that the intermediate samples in the reverse process are drawn from a reward-aware pay-off distribution instead of the reverse process transition distribution. These samples are also used to estimate the gradients during training which avoids the problem of high variance and non-stationarity that arise in classic RL algorithms that use the data generated by the policy to estimate gradients. The pay-off distribution also allows for exploration during reverse diffusion to search for high reward samples. The authors derive a reward-guided loss function based on maximum entropy RL and apply it to diffusion modeling. Experiments on 3D point cloud generation and molecule generation demonstrate that RGDM can effectively generate samples with desired properties as encoded in the reward function.

**Strengths:**

There are several different approaches to guiding the diffusion sampling process based on certain conditions or desiderata. This work proposes, to the best of my knowledge, a novel approach to guided diffusion by adopting ideas from Norouzi et al. (2016). The strength of this approach is that it does not require the reward to be differentiable, and sampling from the pay-off distribution leads to more reliable gradient estimation (as opposed to estimating using policy samples). The paper presents relevant background and then describes RGDM in sufficient detail. The proposed method is fairly simple to understand, and in my opinion, holds moderate significance to the development of a specific sub-area of generative modeling.

**References:**

Norouzi, Mohammad, Samy Bengio, Navdeep Jaitly, Mike Schuster, Yonghui Wu, and Dale Schuurmans. "Reward augmented maximum likelihood for neural structured prediction." *Advances In Neural Information Processing Systems* 29 (2016).

**Weaknesses:**

The main weaknesses are the technical soundness, quality of writing, and the limited empirical evaluation. The derivation of the loss function used for training has several errors, which in the current state make the paper mathematically flawed. The experiments are limited in scope, requiring more categories for the 3D point cloud generation task and more baselines for the molecule generation task. There are several writing and mathematical errors throughout the paper which greatly affect the overall quality. See questions for more details.

In a broader sense, the approach described in the paper is interesting but suffers from one major limitation which is not mentioned anywhere in the paper. Prior methods using classifier-based or classifier-free diffusion can generate different types of samples by changing the conditioning variable during the sampling process. However, RGDM uses the reward function during training, which means the diffusion model can only generate samples tailored to that particular reward function.

**Questions:**

**Derivation in Appendix A.1:** There seem to be several typing errors and some mathematical errors in the derivation of equation 8. Due to the various errors, it is difficult to make sense of certain steps, and the description below is based on some assumptions about the intent.

- Line 2: The expression for differential entropy has a negative sign.
- Line 3: There seems to be a missing integral sign in front of the second term, an unexplained negative sign, and perhaps a missing $\log$ for $q$.
- Line 4: There seems to be no accounting for the feasible set $\mathcal{F}(X_{t-1})$. It is difficult to make sense of this line because of all the errors, especially the missing $\log$.

This derivation forms the basis for the loss function used for training and unless this is clarified, the soundness of the entire approach is brought into question.

**Derivation in Appendix A.2:** The penultimate two steps, which simplify the expression by removing constants is not clear to me. Specifically, $r(X_{t-1})$ is lumped into the constants while performing integral over $X_{t-1}$, which seems incorrect. The argument in the next step "the formulation of $r(X_{t-1})$ is unknown" used to further simplify the expression is also unclear.

**Details on reward-guided sampling:** The last paragraph of Section 4.2 poses the reverse process as a modification of $\tilde{X}_{t-1}$ using reward-conditioned noise for both differentiable and non-differentiable reward functions. This part needs more details into the exact method used to calculate the noise term for both cases, preferably as a formal algorithm. Also, the last sentence of Section 4.3 says "the training process of proposed RGDM is identical to vanilla diffusion models", which contradicts the entirety of Section 4.2 and leads to further confusion.

**Limited experimental evaluation:** For 3D point cloud generation, the experiments only consider one sub-category per class, which seems insufficient to provide a comprehensive empirical investigation of RGDM. For molecule generation, there are several works that could be used as baselines such as Huang et al. (2023), Xu et al. (2023), and Qiang et al. (2023).

**Poor writing and mathematical errors:** There are many spelling, grammatical, and mathematical errors throughout the paper.  I describe some of these here, owing to the lack of line numbers it is a little difficult to specify the precise location. I suggest the authors carefully review the paper from start to finish, and in case LLMs were used, do not rely on their output blindly.

- Citations throughout the paper appear without parentheses. Some citations appear multiple times in the same sentence (for example, in the second paragraph of the introduction Dhariwal & Nicole, 2021, appears four times in two lines). The related works section has some instances of the citation being written in words and then appearing as a citation (for example, "Saharia et al. \cite{Saharia et al., 2021}").
- Instances of strange or incorrect phrasing, such as in the related works section: "a collection of diffusion models", "then perform diffusion models on the latent space", "manipulates the sampling phase of diffusion models to guide a trained model to generate samples", "flexible reinforced framework". Other instances are: using "reversely" instead of "recursively", "nonstation" instead of "non-stationary", "to this point" instead of "at this point".
- The background in Section 3 which describes DDPMs as proposed by Ho et al. (2020) contains mathematical errors. The forward process transitions has mean $\sqrt{\alpha_t}X_{t-1}$ instead of $\alpha_t X_{t-1}$. The equation for the posterior distribution $q(X_{t-1} | X_t, X_0)$ is also incorrect.
- The second term in equation 4 should be in terms of $X_T$ instead of $X_0$.

**References:**

Huang, Lei, Hengtong Zhang, Tingyang Xu, and Ka-Chun Wong. "Mdm: Molecular diffusion model for 3d molecule generation." In *Proceedings of the AAAI Conference on Artificial Intelligence*, vol. 37, no. 4, pp. 5105-5112. 2023.

Xu, Minkai, Alexander S. Powers, Ron O. Dror, Stefano Ermon, and Jure Leskovec. "Geometric latent diffusion models for 3d molecule generation." In *International Conference on Machine Learning*, pp. 38592-38610. PMLR, 2023.

Qiang, Bo, Yuxuan Song, Minkai Xu, Jingjing Gong, Bowen Gao, Hao Zhou, Wei-Ying Ma, and Yanyan Lan. "Coarse-to-fine: a hierarchical diffusion model for molecule generation in 3D." In *International Conference on Machine Learning*, pp. 28277-28299. PMLR, 2023.

Ho, Jonathan, Ajay Jain, and Pieter Abbeel. "Denoising diffusion probabilistic models." *Advances in neural information processing systems* 33 (2020): 6840-6851.

---

### Official Review · Reviewer_X1gM · 2023-11-01

**Soundness:** 2 fair
**Presentation:** 2 fair
**Contribution:** 3 good
**Rating:** 5
**Confidence:** 4

**Summary:**

This paper introduces an innovative diffusion model grounded in reinforcement learning principles, designed to facilitate controllable generation processes. It achieves this by promoting exploration in order to prioritize the generation of highly rewarded samples during the reverse phase. The author adopts an empirical reward mechanism as a guiding principle for diffusion models, and the results demonstrate significant promise across applications in 3D point cloud generation and molecular synthesis.

**Strengths:**

This paper presents a novel perspective by incorporating reinforcement learning concepts into diffusion models, wherein the noise approximation is treated as a Markov multi-step decision process. This innovative approach offers a unique and insightful perspective on the integration of reinforcement learning within the context of diffusion modeling.

**Weaknesses:**

This paper introduces an intriguing approach to training diffusion models. However, the experiments conducted to validate the proposed advantages are somewhat constrained in scope and the content needs to be re-organized.

For experiments:

1. In the point cloud generation experiment, the paper lacks visualized comparisons between various methods, which would have provided valuable insights and clarity in assessing their respective performance.

2. Given that this method is designed for conditional generation, it raises the question of its applicability to other conditional generation tasks, such as image-to-image translation, image inpainting, or text-to-image generation.

3. The author asserts that the model necessitates pre-training, similar to a conventional diffusion model. However, it is worth exploring whether this method could also serve as a post-processing or refinement technique.

4. The experiment exclusively focuses on a three-class generation task, which represents a somewhat constrained scope and is not sufficient to demonstrate its universal advantages comprehensively.

5. The paper lacks baseline comparisons with other types of unconditional generation models, such as auto-regressive models[1], adversarial models [2], flow models [3][5], and energy-based models [4][6].

For the content:
1. Some writing needs to be refined. For example, in the abstract, performances -> performance, i.e.,: -> i.e..

2. The content is not well-organized in this paper. For example, there is an overlap between the introduction and related works for conditional diffusion models. The background of diffusion models in section 3 could be integrated into section 2. The background about 3D point cloud generation could be left in the experiment part as the model is not specific for point cloud generation.

[1] Sun, Yongbin, et al. "Pointgrow: Autoregressively learned point cloud generation with self-attention." Proceedings of the IEEE/CVF Winter Conference on Applications of Computer Vision. 2020.

[2] Arshad, Mohammad Samiul, and William J. Beksi. "A progressive conditional generative adversarial network for generating dense and colored 3D point clouds." 2020 International Conference on 3D Vision (3DV). IEEE, 2020.

[3] Pumarola, Albert, et al. "C-flow: Conditional generative flow models for images and 3d point clouds." Proceedings of the IEEE/CVF Conference on Computer Vision and Pattern Recognition. 2020.

[4] Xie, Jianwen, et al. "Generative pointnet: Deep energy-based learning on unordered point sets for 3d generation, reconstruction and classification." Proceedings of the IEEE/CVF Conference on Computer Vision and Pattern Recognition. 2021.

[5] Shi, Chence, et al. "Graphaf: a flow-based autoregressive model for molecular graph generation." arXiv preprint arXiv:2001.09382 (2020).

[6] Kong, Deqian, et al. "Molecule design by latent space energy-based modeling and gradual distribution shifting." Uncertainty in Artificial Intelligence. PMLR, 2023.

**Questions:**

1. Is there any derivation of Eq(4) for readers without background in RL?
2. What is the performance after pre-training with only MLE?
3. Do the generation objects in Figure 2 come from the same model and different models?